



# Nutrient Flows and Biogeomorphic Feedbacks: Linking Seabird Guano to Plant traits and Morphological Change on Sandy Islands

Floris F. van Rees[1,2,3], Laura L. Govers[1,4], Polina Guseva[2], Maarten P.A. Zwarts[2], Camille Tuijnman[2], Cornelis J. Camphuysen[1], Gerben Ruessink[2], Valérie C. Reijers[2]

[1] Department of Coastal Systems, Royal Netherlands Institute for Sea Research (NIOZ), 't Horntje, 1797 SZ, the Netherlands;
[2] Department of Physical Geography, Utrecht University, Utrecht, 3584 CS, the Netherlands
[3] Department of Ecosystems and Sediment Dynamics, Deltares, Delft, 2629 HV, the Netherlands
[4] Groningen Institute for Evolutionary Life Sciences (GELIFES), University of Groningen, 9700 CC, Groningen, the
Netherlands

*Correspondence to*: Floris F. van Rees (floris.van.rees@nioz.nl)

**Abstract.** Vegetated coastal landscapes are crucial for carbon storage, shoreline protection, and biodiversity. Their structure emerges from biogeomorphic feedbacks between vegetation growth and sedimentation, shaped by environmental conditions. Allochthonous nutrient inputs, particularly seabird guano, can significantly influence plant growth and distribution, potentially

altering these feedbacks. This suggests that coastal birds may actively shape their own habitat by modifying plant-sediment dynamics. Yet, as sea-level rise and coastal squeeze reduce available habitat for already declining bird populations, understanding these interactions becomes increasingly urgent. Despite this, spatially explicit studies on bird–plant–sediment interactions remain lacking. This study addresses that gap by examining how guano deposition influences plant traits, community composition, and landscape morphology. We combined fine-scale field data with remote sensing and spatial

modelling to assess guano effects on vegetation and sedimentation. Field measurements included plant traits, community composition, environmental variables, and $\delta^{15}$N to trace guano uptake. A guano dispersion model was linked to PlanetScope and LiDAR data, and Bayesian models (INLA) revealed spatial links between guano, vegetation change, and sediment accretion. Results show that guano-derived nitrogen promotes shifts in species composition toward later-successional, sediment-stabilizing species, particularly on sandy soils with low baseline nutrient levels. Guano enhanced early-season

vegetation productivity, increasing sediment retention, but seasonal differences and local environmental context modulated these effects. We propose that seabirds act as indirect ecosystem engineers by fueling vegetation–sediment feedbacks. Changes in breeding pair numbers may therefore influence coastal landscape evolution, and ultimately, shape the very habitats these birds depend on.






## 1 Introduction

Vegetated coastal landscapes play critical roles by storing carbon (Macreadie et al., 2019; McLeod et al., 2011), protecting coastlines from erosion and storms (Spalding et al., 2014), and supporting biodiversity (Sutton-Grier & Sandifer, 2019). These coastal landscapes are biogeomorphic systems as they are shaped by reciprocal interactions between biological and physical

processes (Corenblit et al., 2011). For instance, above-ground plant structures modulate wind or water flow, which promotes the trapping of sediments. Similarly, belowground plant structures like roots and rhizomes stabilise sediments by reducing lateral (de Battisti et al. 2019) and topsoil erosion (Marin-Diaz et al., 2021). These biological processes thus enhance surface accretion and shoreline stabilization, which in turn promote vegetation growth. The strength of these biogeomorphic feedback interactions is dependent on the properties and behaviour of above- and below-ground plant structures, commonly referred to

as plant traits. Dune grasses, for example, capture less sand with shorter, thinner or more flexible shoots compared to taller, thicker or stiffer ones (Kuriyama et al., 2005; Van Boxel et al., 1999; Zarnetske et al., 2012).

Plant trait expression is shaped both by species identity and by environmental context. First, different species possess inherently different plant traits. As environmental conditions change, such as through variations in sedimentation, moisture, or salinity, so too can the composition of the vegetation community. In coastal dunes, for example, dynamic beach plains are

typically colonised by burial- and overwash-tolerant pioneer species, while burial-intolerant, sediment-stabilizing shrubs are more common behind developed foredunes (Bakker et al., 2023; Woods & Zinnert, 2024). Because plant traits can influence physical processes, shifts in species composition can have far-reaching consequences for coastal landscape morphodynamics (Schwarz et al., 2018). Second, environmental conditions can also influence intraspecific trait expression. Marram grass, for example, adapts its clonal expansion strategy in response to sedimentation to optimize sand capture (Reijers et al., 2021). In

contrast, other dune- or marsh-forming species express relatively stable clonal traits across a range of environments (Lammers et al., 2023; Ven et al., 2023). Finally, biotic interactions—such as competition, grazing, or facilitation—can further modify plant trait expression, either by shifting community composition or directly altering key ecosystem-engineering plant traits like rooting depth or vegetation height (Kim & Lee, 2022).

In addition to local drivers of environmental conditions and vegetation structure, vegetated coastal ecosystems, often located

along gently sloping gradients, are strongly influenced by cross-ecosystem flows of water, energy, organisms, and nutrients between marine and terrestrial realms. Among these, nutrient flows can particularly profoundly affect plant productivity and trait expression, as well as broader ecosystem properties such as biodiversity, foodweb complexity and stability of food webs (Polis et al., 1997). Mobile consumers play a critical role in these transfers by actively redistributing nutrients across varying timescales, distances, and gradients in ways that abiotic processes alone cannot achieve (Bauer & Hoye, 2014; McInturf et al.,

2019). Examples of these external nutrient pulses to coastal ecosystems include turtle nesting events on sandy shores (Le Gouvello et al., 2017), episodic deposition of washed-up wrack (Joyce et al., 2022), seal feces released during birthing months (McLoughlin et al., 2016) and guano during the breeding season (Benkwitt et al., 2021; Buelow et al., 2018).



Especially guano is an important driver of local ecosystem productivity and biogeomorphic development on various island coastal landscapes, including island atolls (Dunn et al., 2025; Steibl et al., 2024), sandy barrier islands (Reijers et al., 2024)

and mangrove ecosystems (Appoo & Bunbury, 2024). The incorporation of guano is often quantified using stable nitrogen isotopes, with elevated $\delta^{15}N$ levels indicating greater assimilation of nitrogen derived from higher trophic levels (Buelow et al., 2018; Maron et al., 2006; Reijers et al., 2024; Wainright et al., 1998). Overall, guano fosters vegetation growth (Anderson & Polis, 1999; Barrett et al., 2005; J. Ellis, 2005; Maron et al., 2006; Young et al., 2011). Therefore, biogeomorphic effects can be positive when guano-derived nutrients increase above and belowground biomass productivity, as this enhances sediment

capture and erosion resistance (Dickey et al., 2023; Morton et al., 2025). On the other hand, effects can also be negative when fertilization by guano reduces the investment in belowground root structures, which affect erosion resistance (Marin-Diaz et al., 2021; Pavlik, 1983). So far, most studies assess the effects of bird guano on whole island ecosystem productivity by extrapolating findings on a few square meters to the whole island scale (Anderson et al., 2008; Ellis, 2005; Reijers et al., 2024), whereas local environmental conditions, such as differences in landscape morphology and hydrodynamic exposure, and spatial

gradients in nutrient subsidy might differ strongly. This means that the fertilizing impact of guano is likely underestimated when averaged across whole-island scales. In reality, its effects are spatially heterogeneous, varying in both magnitude and direction, depending on local conditions such as topography, hydrodynamics, and nutrient deposition gradients. These spatial differences can substantially shape biogeomorphic processes, influencing vegetation dynamics, sediment capture, and ultimately, the formation of bird habitat. As coastal ecosystems face mounting pressures from sea-level rise (van de Pol et al.,

2024) and coastal squeeze (Lansu et al., 2024), understanding how guano alters habitat structure at fine spatial scales is critical for the conservation of already declining bird populations (Paleczny et al., 2015).

Our study aims to disentangle the reciprocal relationships between guano, vegetation composition and traits, environmental conditions, and morphological changes on sandy biogeomorphic islands. To understand the role of bird-plant interactions in soft sediment coastal ecosystems, small uninhabited barrier islands are ideal living laboratories because of their dynamic

nature. Combined with limited human interference, these islands create unique ecological niches that are particularly favourable for seabird populations (Foster et al., 2009). We applied a multi-variable framework on two spatial levels using the Dutch Wadden Sea model system. First, we characterised the (in)direct interactions between vegetation composition, plant trait expression, and the environmental conditions on five sandy islands. This was conducted on a fine spatial resolution (4 m$^2$) with limited spatial coverage using structural equation modelling that allows for integrating both direct and indirect effects,

including spatial dependency (Lefcheck, 2016). We hypothesize that plant trait expressions are influenced both directly and indirectly by guano deposition and other abiotic factors. Secondly, we applied remote sensing techniques to investigate the interaction between guano deposition, plant productivity and landscape morphological changes. We hypothesize that guano promotes vegetation growth, thereby enhancing vegetation-driven sediment capture. This approach was performed on a coarser spatial resolution (9 m$^2$) with a full island coverage using a Bayesian hierarchical joint modelling approach with an integrated

nested Laplace approximation to include a spatial correlation structure. Integrating fine-scale in situ monitoring with broad-



scale remote sensing provides a mechanistic understanding of biogeomorphic feedbacks (Cavender-bares et al., 2022; Lausch et al., 2018) and elucidates the net effects of seabird guano on island morphology.



## 2 Methods

### 2.1 Sites description

Our study was performed on five inhabited sandy islands situated in the Dutch Wadden Sea region, Rottumeroog (53°32'25"N, 6°34'55"E), Rottumerplaat (53°32'30"N, 6°28'51"E), Richel (53°17'50"N, 5°8'5"E), Griend (53°15'55"N, 5°15'15"E), and Zuiderduin (53°31'0"N, 6°35'0"E), (Figure 1). The Wadden Sea consists of vast tidal flats shielded by chains of barrier islands stretching 500 km from the Netherlands to Denmark (Kabat et al., 2012). The global significance of this region for millions of migratory birds has led to the Wadden Sea being designated as a UNESCO World Heritage Site. The tidal flats harbor a diverse community of benthic organisms (Compton et al., 2013), making it a vital foraging habitat for shorebirds (Boere & Piersma, 2012).

All islands are uninhabited, protected nature reserves that remain closed to the public year-round. They were selected because each island is sandy in origin, is shaped by feedback interactions between vegetation growth and sedimentation processes that drive their morphodynamic development, and harbours coastal bird colonies. Rottumeroog and Rottumerplaat are classic barrier islands, formed under wave-dominated conditions and shaped by aeolian processes that promote dune formation, which were affected by the construction of a sand drift dike around 1950. In contrast, Richel originated as an unvegetated sand shoal within the flood tidal delta. Since 2009, it has rapidly evolved into a vegetated sandy island, with dune development primarily driven by aeolian processes enabled by its long fetch. Zuiderduin separated from Rottumeroog around 1930 and has developed into a sandy back-barrier island (Reijers et al., 2024). Finally, Griend has existed since the Middle Ages but extensive erosion after 20th-century closure dam construction led to the construction of artificial stabilization of the island between 1980-1988. A 2016 sand nourishment (~200000 m³) was added to protect its critical breeding habitat, using locally dredged, nutrient-poor sand (Reijers et al., 2024).

The number of different breeding pairs and the number of species differ markedly between the islands, reflecting the varying habitat conditions and management histories (S1 in the supplements). *Larus argentatus* (Herring Gull) and *Larus fuscus* (Lesser Black-backed Gull) dominate on all islands but are especially numerous on Rottumerplaat and Griend. *Chroicocephalus ridibundus* (Black-headed Gull) is only present on Zuiderduin and Griend, with a striking concentration on Griend. *Phalacrocorax carbo* (Great Cormorant) breeds on Zuiderduin and Richel, while *Larus canus* (Common Gull) is limited to Zuiderduin and Rottumeroog. Notably, Griend hosts the highest species diversity, including large colonies of *Thalasseus sandvicensis* (Sandwich Tern), *Platalea leucorodia* (Eurasian Spoonbill), *Sterna paradisaea* (Arctic Tern), and *Sterna hirundo* (Common Tern), which are absent from the other islands.





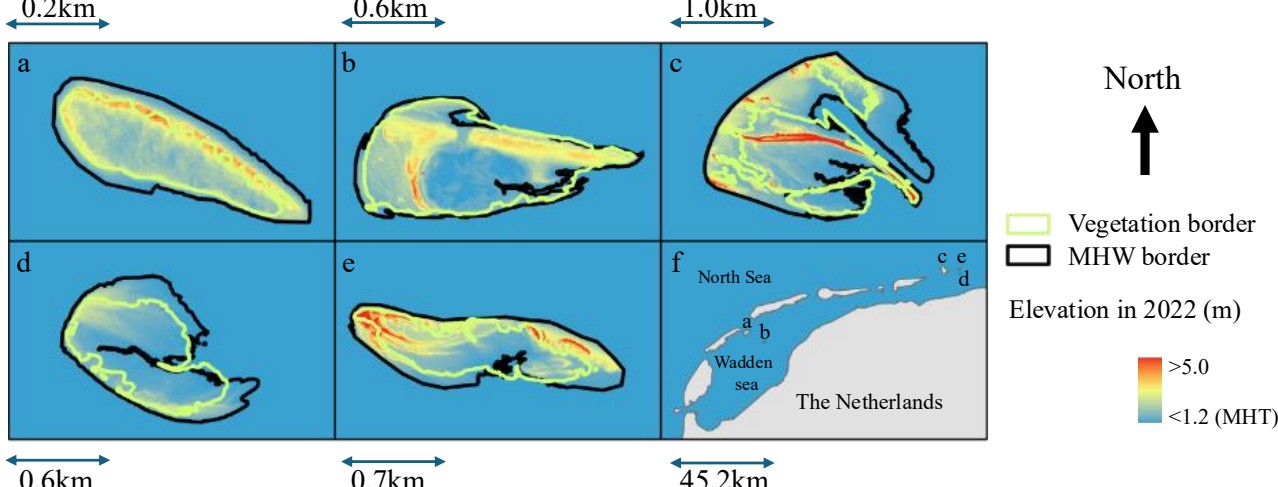

**Figure 1: The five islands included in this study: Richel (a), Griend (b), Rottumerplaat (c), Zuiderduin (e), and Rottumeroog (e). Their relative location with respect to the mainland is displayed in panel (f). The colormap, elevation in 2022, shows the elevation**
**with respect to the Dutch ordinance datum (NAP), based on LiDAR pointclouds *(Rijkswaterstaat, 2017)*. The border of the vegetation is indicated by a green line defined as an NDVI higher than 0.2, as used in (Reijers et al., 2024), and the border of mean high tide (MHW) is depicted by a black line.**

## 2.2 Study set-up

We used a two-tiered approach to examine how guano affects vegetation growth and sedimentation. First, we analysed how
guano-derived nitrogen influences vegetation composition and plant traits at the plot level (4 m²) across five islands using structural equation models and field data. Second, we scaled up to the island level by linking guano deposition to satellite-derived vegetation indices (NDVI, GI) and LiDAR-based elevation change (ΔZ) from 2021 to 2022. This framework connects guano input to broader landscape changes in vegetation and elevation. An overview of the methodology is provided in Figure 2.



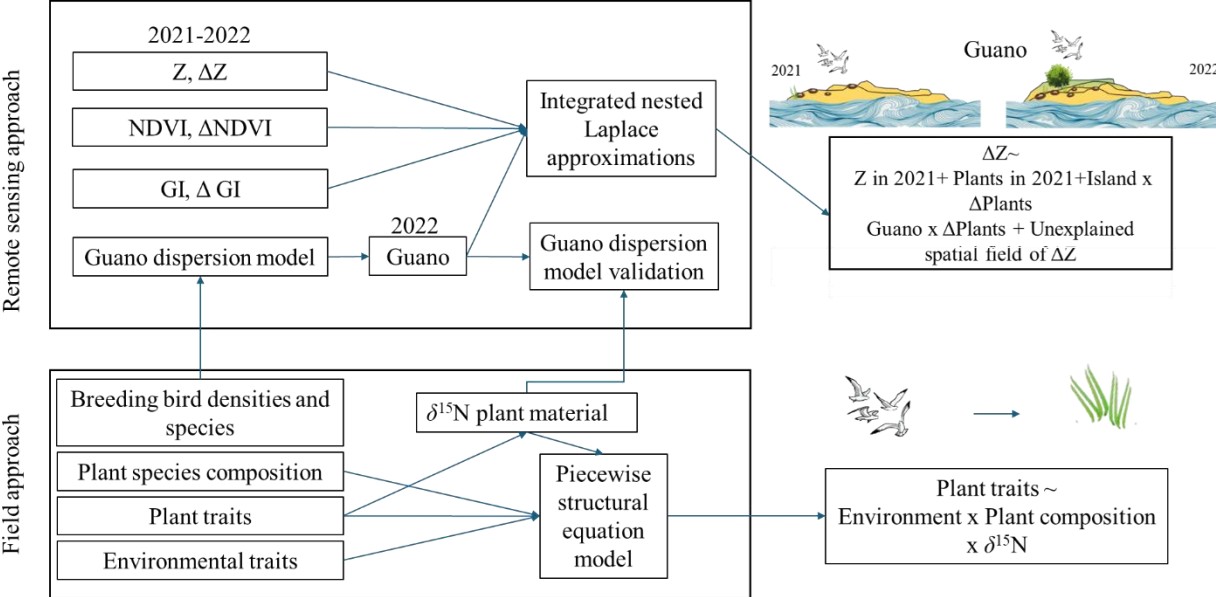

**Figure 2: Schematic overview of the two-tiered methodological framework. Field approach: Structural equation modelling of how plant traits are driven by species composition, environmental conditions and foliar δ¹⁵N (as a proxy for seabird-derived guano). Remote sensing approach: Island-wide Bayesian spatial modelling (INLA) of elevation change (ΔZ) in relation to vegetation state (NDVI, GI), its temporal changes (ΔNDVI, ΔGI) and guano deposition, with island-specific effects and spatial autocorrelation accounted for.**

## 2.3 The effect of guano deposition on species composition and plant traits

On all five islands, we sampled vegetation composition, environmental conditions, and plant trait expression along transects ranging from 300 to 700 meters in length (5–6 sample plots per transect), resulting in 24-30 plots per island, 118 in total. Transect placement was designed to capture variation in guano deposition, spanning both within and outside bird colonies. As a result, transect lengths varied depending on the spatial configuration of colonies and the size of each island. Richel and Rottumeroog were sampled haphazardly as the island width was too short to accommodate transect sampling. Sampling was conducted in August and September 2022, shortly after the breeding season. At each sampling plot, we identified plant species and estimated their percent cover within 4 m² plots to characterized the vegetation community composition. Vegetation was clipped within a 0.4 × 0.4 m frame placed in a representative area to determine plant biomass of the total community (g DW m$^{-2}$). Vegetation biomass was then determined after drying the samples at 60°C for 48 hours. Rooting depth was determined as the average of two 1 m depth soil profiles (∅ 2.5 cm), from which we measured the distance between the surface and the deepest living root. Vegetation height of the community was determined as the average of four measurements taken within a 10 cm radius of plot corners. Within each plot, we collected approximately five leaves from the three most abundant plant species to measure foliar carbon and nitrogen levels and the isotopic ratio of nitrogen (δ¹⁵N), which serves as a proxy for guano



assimilation. Elevated $\delta^{15}N$ values ($\delta^{15}N > 10$) indicate that plants have incorporated organic nitrogen derived from guano (Maron et al., 2006; Reijers et al., 2024). From the three species available per plot, we picked the species that was most common in all plots to decrease the number of different plant species measured. This was done to minimize the influence of species-specific nitrogen isotope fractionation. If the most common species was absent from a plot, the next most frequent species was selected, and this process was repeated as needed.

The plant samples for elemental analysis were initially rinsed with demineralized water, freeze-dried, and then finely ground and homogenized using a CG-200 Freezer/Mill Compact Cryogenic Grinder. Approximately 1 mg of each homogenized sample was placed into tin cups. These samples were analysed for $\delta^{15}N$ isotope ratios, using a Carlo-Erba NA-1500 Elemental Analyser coupled with a Thermo Finnigan DeltaPlus mass spectrometer via a Finnigan ConFlo III Universal Interface.

We also sampled environmental variables in each sampling plot. Included in our analysis were 1) soil organic matter content,

2) elevation relative to the Dutch ordnance datum (NAP), which is about mean sea level; and 3) distance from the coast. For soil organic matter, we collected sediment samples (100 mL) from the top 5 cm of soil. The organic matter content was then determined as loss on ignition by combusting dry sediment samples at 575°C for four hours (Heiri et al., 2001). Elevation was measured using a Topcon HiPer SR single RTK-GPS system. The distance from the coast was calculated as the Euclidean distance between each sampling point and the nearest cell in a raster indicating areas below mean high water in 2022. This

elevation raster was derived from a LiDAR point cloud of the Dutch coast (Rijkswaterstaat, 2017).

To investigate how guano deposition affects the reciprocal relationships between environmental conditions, vegetation composition, and plant trait expression, we applied structural equation modelling (SEM) in R using the piecewiseSEM package (version 4.3.2). We used ordination techniques to reduce the dimensionality and capture key gradients into main axes for both the environmental variables, plant traits and vegetation community composition. For the environmental variables and the plant

traits, we applied principal component analyses. We retained the principal components axes that explained at least 80% of the variance for both environmental conditions and plant trait data, and these components were subsequently incorporated into the SEM models (Zhang et al., 2024). Environmental conditions were represented by two principal component axes (88.4% variance, S2 in the supplements), while plant traits were summarized by three axes (82.7% variance, S3 in the supplements). Principal components were computed using singular value decomposition (SVD) via the prcomp() function in R, which ensures

numerical stability and accurate handling of correlated variables. Instead, non-metric multidimensional scaling (NMDS) is preferred to reduce dimensionality for compositional data such as species cover because it accommodates non-Euclidean distances, is robust to zero inflation, and better captures ecological dissimilarity patterns (Aerts et al., 2006). Therefore, we applied NMDS using the *vegan* package in R (version 2.6-4). The first two NMDS axes (NMDS1 and NMDS2), which captured the major gradients in species composition, were included in the SEM to represent plant community composition in

accordance with Kahmen et al. (2005).

To quantify the effect of guano on vegetation community composition and trait expression we constructed a SEM that excluded the effect of guano, including only environmental conditions as a driver of vegetation characteristics, and compared this null model to the full model that included foliar $\delta^{15}N$ values as proxy for guano deposition (Figure 3 for the hypothesized




relationships). This model included all hypothesized pathways and was refined through backward elimination, removing

pathways with p > 0.05. Shipley's test of directional separation (D-separation) was employed to ensure that previously omitted significant relationships were included, improving model fit (Shipley, 2009).

Spatial autocorrelation in residuals was addressed using spatial lag models (lagsarlm function from the spatialreg package, version 1.3-2), with spatial relationships defined by k-nearest neighbor networks (k = 4). Residuals were tested for spatial dependence using Moran's I (spdep package, version 1.3-1), and spatial adjustments were iteratively applied until no

significant autocorrelation remained. Individual pathways were evaluated for normality using the Shapiro-Wilk test on residuals and assessed for homoscedasticity through residuals versus fitted value plots.

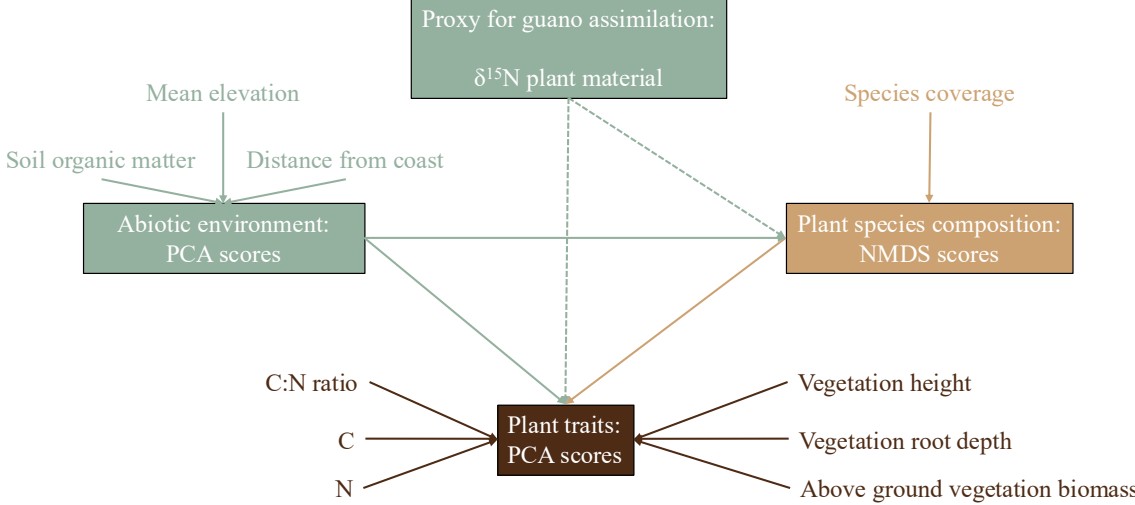

**Figure 3: Design for the SEM in which we hypothesize that environmental conditions, represented by PCA, determine plant species composition, represented by NMDS scores. Both environmental conditions and plant species composition influence plant trait**
**expression. Additionally, guano assimilation, approximated by δ¹⁵N, acts as an isolated environmental factor that directly affects both plant species composition and trait expression..**

## 2.4 The effect of guano deposition on vegetation-mediated morphological changes

To explore how guano deposition affects island landscape-scale biogeomorphic processes, we used spatial data on the presence of breeding nests to estimate guano deposition, satellite imagery to calculate vegetation productivity indexes and lidar-derived

coastal elevation models to explore bed elevation changes.

During the 2022 breeding season, bird wardens recorded the location and density of nesting birds. These data were used to create spatial polygons of nest density per species. Guano deposition was estimated using species-specific fecal excretion rates derived from bird body mass, diet, and metabolic efficiency, based on established allometric equations (Hahn et al., 2007; Karasov, 1990; Nagy et al., 1999), building on the methodology described in Reijers et al., (2024). Bird biomass was



determined by linking bird species ID to the global trait database Tobias et al. (2022). A daily excretion rate was calculated and multiplied by nest density and breeding duration to estimate total guano deposition per area per year.

To model the spatial distribution of guano, deposition was assumed to decrease linearly with distance from the nest (Bokhorst et al., 2019; Savage, 2019), up to a maximum of 300 meters (Benkwitt et al., 2021). We refer to this range as the dispersion length throughout this study. A 1-meter resolution Euclidean distance matrix was created around each colony. This matrix was

used to generate a distance-decay function, scaling the guano deposition accordingly with its dispersion length. To maintain mass balance, deposition values within colonies were iteratively reduced until the total guano within the colony matched the sum of dispersed deposition in the colony and the surrounding area.

In cases of overlapping colony effects, guano contributions were summed per cell. To evaluate how far guano affects nitrogen assimilation, the model tested dispersion lengths ranging from 1 to 300 meters. Theoretical deposition was then compared with

measured foliar $\delta^{15}N$ values from the field, and $R^2$ values were used to determine which spatial scale best predicted nitrogen enrichment. Comparison of the guano dispersion model and foliar $\delta^{15}N$ values from sampled plants showed that a dispersion length of 300 m proved the best fit ($R^2$: 0.11, S6 in the supplements).

### 2.4.1 Remote sensing of plants and LiDAR for elevation

To study the effects of guano deposition on biogeomorphic interactions, we assessed the impact of guano-induced vegetation

changes on sediment bed level dynamics. Vegetation state was represented using the Normalized Difference Vegetation Index (NDVI), calculated from PlanetScope satellite imagery at a 3-meter resolution. The minimum and maximum number of pixels per island were 10,979 (Richel) and 501,266 (Rottumerplaat) (S4 for island specifications). The imagery consisted of orthorectified GeoTIFF files with four spectral bands (near-infrared, red, green, and blue), which compensate for terrain distortions and sensor artifacts to provide Top of Atmosphere (TOA) radiance. NDVI was calculated using the red (R) and

near-infrared (NIR) spectra (Tucker, 1979): $NDVI = (NIR - R)/(NIR + R)$.

PlanetScope employs multiple satellites, so to minimize both instrumental and temporal variability, we computed two separate vegetation metrics from different seasonal subsets of the imagery. First, we calculated the mean NDVI from all cloud-free images acquired during the summer months (June–September). This summer mean was used as the representative NDVI value for each year. Second, because the breeding season starts in spring and can influence early vegetation growth, we calculated a

Greening Index (GI) based on all NDVI images from the spring period (March–May). GI was defined as the slope of a linear model fitted through the springtime NDVI values, indicating the rate of NDVI change over days.

Elevation data was derived from the LiDAR dataset of the Dutch coast (Rijkswaterstaat, 2017), and converted to a 3-meter resolution raster. Vegetation dynamics were quantified as year-to-year changes in GI and NDVI between 2021 and 2022:



$\Delta GI = GI_{2022} - GI_{2021}$ and $\Delta NDVI = NDVI_{2022} - NDVI_{2021}$ respectively. These were analysed alongside elevation change

over the same period: $\Delta Z = Z_{2022} - Z_{2021}$.

### 2.4.2 Modelling Framework

We modelled sediment bed level change ($\Delta Z$) between 2021 and 2022 across five Wadden Sea islands as a function of vegetation dynamics, guano deposition, and elevation. Vegetation growth can promote sediment accretion, while its decline may lead to erosion. These vegetation changes are hypothesized to be influenced by guano fertilization and may vary by island.

To test these relationships, we developed two Bayesian spatial models because potentially guano deposition can change the rate of greening in spring, and/or the resultant biomass after the breeding season in summer. Both models include interaction terms between guano deposition and vegetation dynamics, and between vegetation dynamics and island identity. They also account for the initial vegetation state and elevation in 2021:

    **ΔGI model** uses $\Delta GI$ to represent the difference in greening rate in spring between 2021 and 2022.

**ΔNDVI model** uses $\Delta NDVI$ to represent changes in vegetation biomass in summer between 2021 and 2022.

To account for spatial autocorrelation in elevation change, we included a spatially structured random field (RF) in both models. This field was modelled using the Stochastic Partial Differential Equation (SPDE) approach in INLA, which approximates a Gaussian Random Field through a sparse precision matrix defined by a Matérn covariance function (Bakka et al., 2018; Zuur & Ieno, 2018). We applied Penalized Complexity (PC) priors to control spatial smoothness: a prior median range of 300 m

(Pr(range > 300 m) = 0.5), corresponding to the size of the smallest island (Richel), and a PC prior on the marginal standard deviation (Pr($\sigma$ > 1) = 0.1), allowing sufficient flexibility while regularizing spatial variation.

The model structures are:

    **Model 1:**

$$\Delta Z = \beta_1 Guano \; x \; \Delta GI + \beta_2 GI_{2021} + \beta_3 Z_{2021} + \beta_4 island \; x \; \Delta GI + RF$$

**Model 2:**

$$\Delta Z = \beta_1 Guano \; x \; \Delta NDVI + \beta_2 NDVI_{2021} + \beta_3 Z_{2021} + \beta_4 island \; x \; \Delta NDVI + RF$$

We then subsampled up to 1000 vegetated cells per island to balance spatial coverage and computational load. Cells were classified as vegetated if NDVI > 0.2, following Reijers et al. (2024). Guano deposition was log-transformed as log(Guano + 1) to reduce the influence of extreme values. All continuous variables were standardised by subtracting the mean and dividing

by the standard deviation.

A triangular mesh was created for each island to define the spatial domain, with a maximum edge length of 30 m within islands and up to 3000 m between islands. The mesh boundary followed the MHT line of the islands. Models were fitted in R using the INLA package, which provided estimates for fixed effects, posterior spatial fields, and model diagnostics (DIC and WAIC).





Residuals were evaluated for normality, homoscedasticity, and spatial structure. Key covariates and interaction terms were
visualized against residuals to validate model assumptions.

## 3 Results

### 3.1 The effect of guano deposition on species composition and plant traits

3.1.1 Reduced dimensionality of environmental variables, plant species composition, and plant traits

To evaluate how environmental conditions influence vegetation traits and composition, we constructed a piecewise structural
equation model (SEM) (Figure 5). Higher values on $PC1_{environment}$ represent inland sites with higher organic matter content.
$PC2_{environment}$ increases toward lower-elevation sites (Figure 4a, and S2 in the supplements). PCA scores related to plant traits
are displayed in Figure 4c-d; loadings can be found in S3 in the supplements. Higher $PC1_{plant\ trait}$ scores reflect lower vegetation
height, biomass, root depth and foliar carbon. $PC2_{plant\ trait}$ increases with higher nitrogen content and lower C:N ratio. $PC3_{plant\ trait}$ captures a trade-off between rooting depth and aboveground height: higher $PC3_{plant\ trait}$ scores indicate shallow-rooted but
taller plant.

Species composition was ordinated using NMDS (stress = 0.13). Higher NMDS1 values indicate communities dominated by
sand-preferring species, while lower values reflect mud-preferring species (Figure 4b). Along NMDS2, shifts depend on
substrate origin: on sand, increasing values correspond to grasses like *Ammophila arenaria*, while decreasing values reflect
more forbs such as *Atriplex littoralis*. In muddy environments, increasing NMDS2 values indicate a shift toward pioneer
species such as *Salicornia procumbens*, while decreasing values reflect a transition toward climax species like *Elytrigia
atherica*.





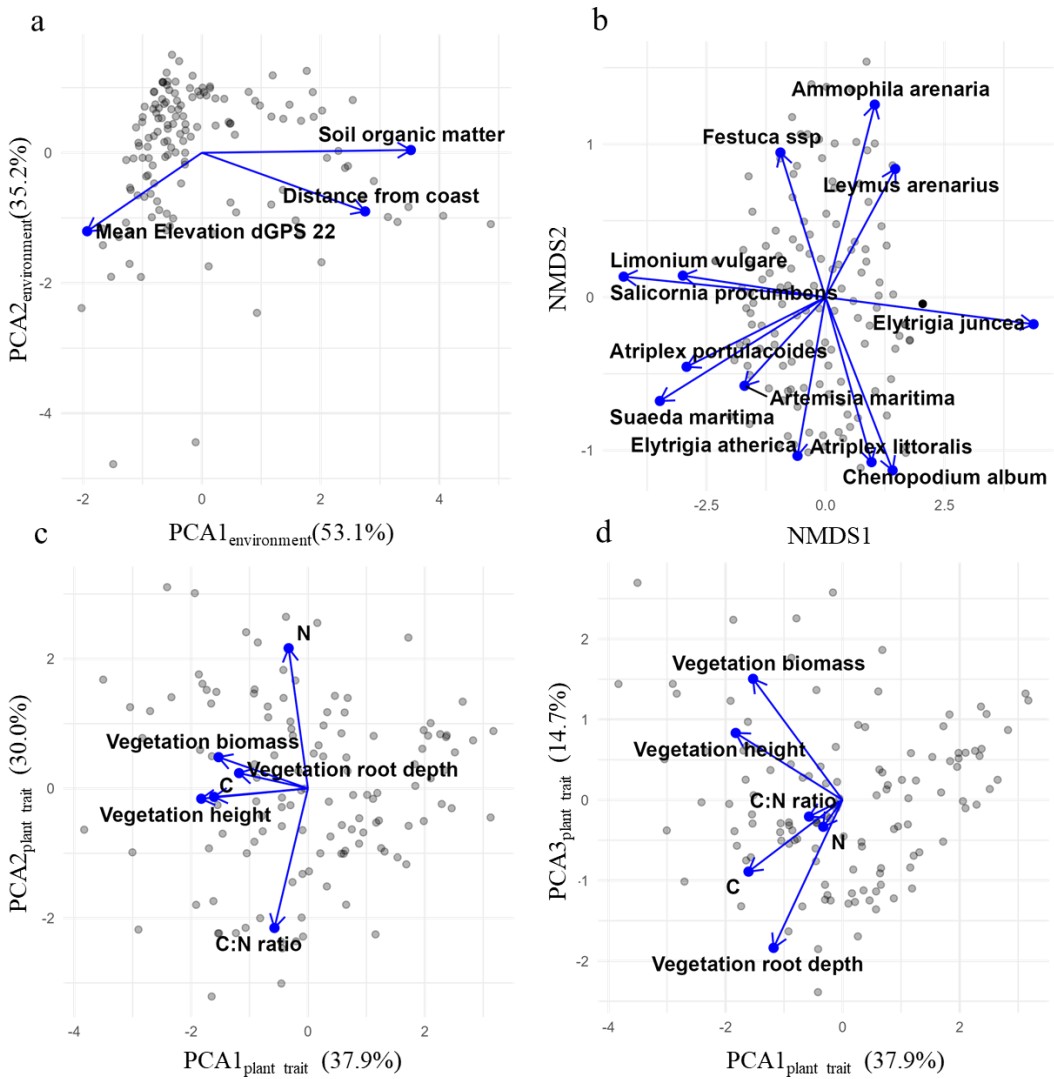

**Figure 4: PCA and NMDS bi-plots of (a) environmental variables, (b) plant species composition, and (c-d) )plant traits. The NMDS plot displays the 12 most abundant species.**

3.1.2 The (in)direct effects on environmental conditions on vegetation composition and trait expression

In the model without $\delta^{15}N$, organic-rich soils are associated with mud-preferring species (PC1$_{environment}$ → NMDS1: –0.32), and lower elevations also shift composition toward mud-preferring communities (PC2$_{environment}$ of environment → NMDS1: –0.30), (Figure 5b). These compositional shifts affect trait expression: sand-preferring species generally have higher biomass and deeper roots (NMDS1 → PC1$_{plant\ trait}$: –0.31), while forbs on sand or climax species on mud have higher nitrogen and lower C:N ratios (NMDS2 → PC2$_{plant\ trait}$: –0.33), (Figure 5c). Elevation directly promotes traits linked to higher biomass and deeper roots (PC2$_{environment}$ → PCA1 of plant traits: 0.23). This indicates that within the same species composition, these traits can be



altered by elevation differences. The model without $\delta^{15}$N explains 67% of NMDS1, 21% of NMDS2 and 44% of PC1$_{\text{plant trait}}$, 20% of PC2$_{\text{plant trait}}$ and 27% of PC3$_{\text{plant trait}}$, which changes upon the inclusion of $\delta^{15}$N.

When we include $\delta^{15}$N, the model explains 9% more variance of plant species composition (NMDS2), and 5% more variance
of plant traits (PC2$_{\text{plant trait}}$). This updated SEM shows that guano assimilation shifts composition toward climax in mud or forb-dominated communities in sand ($\delta^{15}$N → NMDS2: –0.32) and increases foliar nitrogen content independent of species composition ($\delta^{15}$N → PC2$_{\text{plant trait}}$: 0.36), (Figure 5c and S5 in the supplements). Because NMDS2 feeds into PC2$_{\text{plant trait}}$, $\delta^{15}$N also indirectly influences plant trait through changing the plant species composition. Additionally, a new path emerges in which lower elevation is associated with lower nitrogen concentrations (PC2$_{\text{environment}}$ → PC2$_{\text{plant trait}}$: 0.17). The latter
compensates for the fact that higher foliar $\delta^{15}$N is associated with higher N; this effect is reduced in low-lying areas like salt marshes, likely because $\delta^{15}$N is less assimilated by salt marsh species, because of the abundance of organic matter (S5 in the Supplements).



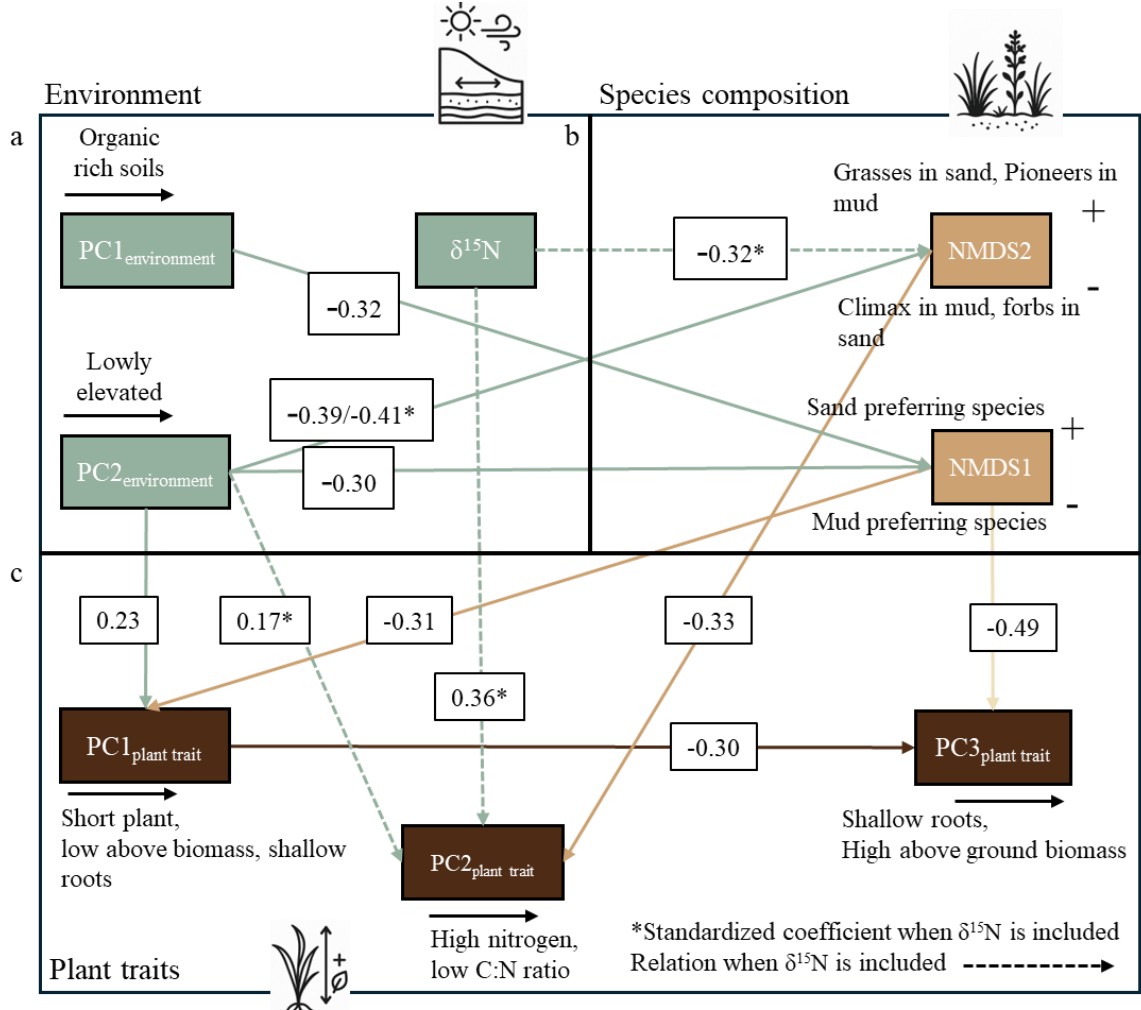

**Figure 5: Piecewise SEM of (a) environmental PCA scores, (b) species-composition NMDS axes, and (c) plant-trait PCA scores.**
**Arrows show significant pathways with standardized coefficients; an asterisk (\*) marks coefficients from the $\delta^{15}$N-inclusive model**
**($\delta^{15}$N as guano proxy), and dashed arrows are new $\delta^{15}$N-driven paths. Goodness-of-fit statistics of the model without $\delta^{15}$N indicate**
**adequate model performance ($\chi^2$ = 14.86, p = 0.25; Fisher's C = 30.62, p = 0.17; AIC = 1478). Upon the inclusion of $\delta^{15}$N, the updated**
**model displays an improved fit($\chi^2$ = 20.15, p = 0.17; Fisher's C = 40.21, p = 0.10; AIC = 1459).**

## 3.2 The effect of guano deposition on vegetation-mediated morphological changes

### 3.2.1 Vegetation state changes by guano deposition, affecting morphological changes

Two Bayesian hierarchical modelling approaches enabled us to analyse how the timing of vegetation development and
fertilization by guano affect sediment trapping through shifts in vegetation state (Figure 6, and S7-10 in the Supplements).
This effect is analysed through the interaction between guano, changes in vegetation state and change in sediment bed level.



We found that the interaction between guano and vegetation development differed notably between models. In the ΔGI model,
the interaction between guano and ΔGI was positive (0.010 ± 0.005 SD), indicating that guano fertilization enhances the ability
of spring vegetation greening to promote sedimentation, in line with the hypothesis (Figure 6). Surprisingly, the ΔNDVI model,
the interaction between guano and ΔNDVI was negative (-0.033 ± 0.010 SD) (Figure 6). This means that positive bird-induced
sediment bed level change is expected at locations that are high in guano deposition, and negative in ΔNDVI.

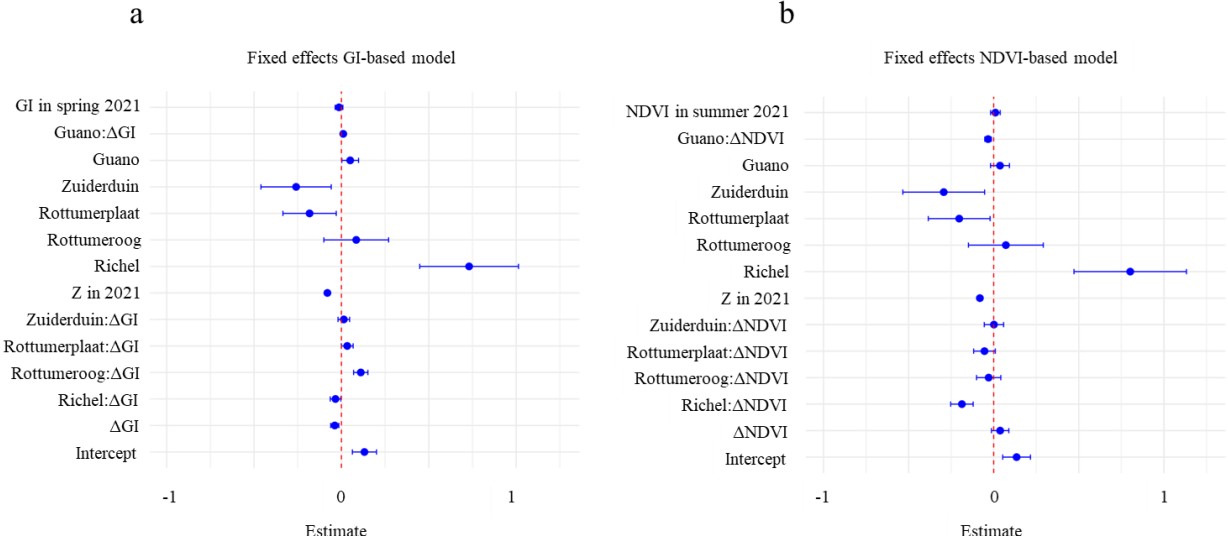

**Figure 6: Fixed-effect estimates (markers) with 95% credible intervals for (a) GI-based and (b) NDVI-based spatial models. All**
**predictors were standardized. The GI model's spatial range (175 m) is shorter than the NDVI model's (206 m), and both have similar**
**random-field SD (~0.3 m). Model DIC/WAIC are comparable (GI: −2203/−2295; NDVI: −2228/−2290). Spatial maps of ΔGI, ΔNDVI**
**and ΔZ are in Supplement S9–11.**

### 3.2.2 The importance of the guano-induced changes in sediment bed level change varies spatially

The impact of guano deposition on an island's morphology depends on the relative influence of other factors shaping sediment
dynamics. The spatially explicit percentual contribution to elevation change is depicted in Figure 7. On islands like
Rottumeroog and Rottumerplaat, where guano deposition is low and vegetation remains stable, but morphological change is
significant, our models attribute this variation primarily to island-specific effects or the spatial random field. Conversely, on
islands where guano deposition is high but the inherent depositional trend is weak, guano-driven sedimentation plays a more
prominent role, as observed on the south of Griend and west of Zuiderduin, where sedimentation is explained up to 20% by
interactions between guano and vegetation. In contrast, Richel also experiences high guano deposition, but its naturally high
sedimentation rates dominate morphological change, reducing the relative contribution of guano in shaping its landscape.



**Figure 7: Predicted proportion (%) of change in elevation imposed by an interaction effect between the change in vegetation state (GI or NDVI) and guano deposition wrt the absolute value of all modelled terms. The interaction term can become negative, depending on the sign of the vegetation state change. The predicted proportion of change in elevation, can therefore, also be negative. Negative means erosion, positive is erosion wrt the absolute values of all terms of the prediction. The thick black lines indicate the edge of the island, the MHT line. The thin black lines indicates the location of the colonies.**





### 3.2.3 Birds breed at stable sites

Guano deposition, without interaction, had a positive effect on changes in sediment bed level in both models (Figure 6). From an ecological perspective, this can be viewed as a positive association between breeding bird densities and locations where less erosion occurs. In the ΔGI model, guano deposition had an effect of 0.050 ± 0.024 SD on changes in sediment bed level. In the ΔNDVI model, the effect was slightly lower (0.037 ± 0.028 SD), suggesting a weaker direct relationship between guano and morphological change when approximating vegetation through ΔNDVI.

**3.2.4** Abiotic and island-specific contributions to sediment dynamics

While the direct effects of guano deposition on sediment bed level change are central to this study, both models show that other factors also play a role. Initial elevation in 2021 was a strong negative predictor of sediment bed level change (GI-based: -0.080 ± 0.007 SD; NDVI-based: -0.081 ± 0.007 SD), indicating that lower areas were more prone to accretion or less erosion. Island identity also significantly influenced geomorphic responses: Richel showed more accretion (GI-based: 0.731 ± 0.144
SD; NDVI-based: 0.802 ± 0.169 SD), while Rottumerplaat and Zuiderduin showed more erosion compared to Griend. Interestingly, several islands exhibited negative interactions between vegetation change and sedimentation, particularly Richel (GI-based: -0.034 ± 0.015 SD; NDVI-based: -0.187 ± 0.034 SD), where vegetation decline may reflect rapid burial due to strong sedimentation.



## 4 Discussion

Our study demonstrates that seabird guano is a biotic driver modulating vegetation dynamics and, in turn, biogeomorphic interactions. While classic environmental filters such as elevation and soil organic matter remain primary determinants of vegetation composition and functional trait expression, guano shifts plant communities toward more nitrophilous, high-productivity species and increases foliar nitrogen, as shown by our structural equation models (Figure 5). However, we found no consistent evidence that guano alters above- or belowground traits typically associated with sediment stabilization. At the landscape scale, elevation was the dominant predictor of elevation change, with lower-lying areas experiencing more erosion or less accretion in the years we analysed. Guano- or vegetation-mediated effects on elevation were generally weak, except in localized areas where their interaction explained up to 22% of variation in sediment dynamics. This spatial heterogeneity was also reflected in island-specific patterns: at Rottumeroog, early-season greening had a strong positive effect on sediment accretion, whereas at Richel, high background sedimentation appeared to mask the influence of vegetation. Together, these findings suggest that while guano influences vegetation composition and traits related to nitrogen assimilation, its role in modulating vegetation–landform feedbacks is highly context-dependent. Integrating trait data from the field with landscape-scale remote sensing thus provides a powerful approach to unravel the complex, spatially variable effects of biotic nutrient inputs on coastal landscape development (Lausch et al. 2018; Cavender-Bares et al. 2022).

### 4.1 Guano can change local vegetation characteristics

Our findings demonstrate that guano can drive shifts in plant species composition, which, in turn, affects trait expression. The inclusion of $\delta^{15}N$ in our structural equation models revealed an improved explanatory power, explaining 9% more variance in plant species composition and 5% more variance in plant traits, accentuating that guano overrides or modifies abiotic constraints (Figure 5). These results support the conceptual framework of bottom-up, nutrient-driven succession in coastal systems (Olff et al., 1993). Specifically, nutrient-rich guano promotes the proliferation of nitrophilous, later-successional species on both sand (e.g., *Atriplex littoralis*) and mud (e.g., *Elytrigia atherica*), independent of background community structure. This is in correspondence with earlier studies on *Atriplex ssp.* on seabird islands (Anderson & Polis, 1999). Yet, plants in nutrient-deprived sandy soils appear to be more reliant on guano subsidies, as we observed that primarily sand-preferring communities displayed foliar $\delta^{15}N$ enrichment, suggesting elevated nitrogen assimilation from guano (S5 in the Supplementary Material). These fertilization effects are further reflected in elevated foliar nitrogen and reduced C:N ratios, particularly in forb-dominated or climax communities, comparable with earlier observations (McKane et al., 2016; Reijers et al., 2024; Wainright et al., 1998). Sand-preferring species also exhibit traits linked to resource acquisition, such as greater biomass and deeper rooting. Root growth is generally enhanced in nitrogen-limited soils (Ericsson, 1995), such as sand, in comparison to mud.

Additionally, even within a given species composition, plants show substantial trait plasticity in response to elevation (Schulte Ostermann et al., 2021). In lower-lying sites, characterised by more saline or frequently inundated conditions, plants tend to




develop shallower roots, lower biomass, and reduced vegetation height, potentially as adaptive strategies to cope with environmental stress (Battisti et al., 2020). However, despite clear evidence that guano influences species composition and nutrient assimilation, we did not find a consistent link between guano enrichment and the strengthening of biogeomorphological traits like above-ground biomass, vegetation height, or rooting depth. This finding is partly in agreement with Reijers et al. (2024), who also observed guano-induced nitrogen assimilation. However, unlike our results, they identified a quadratic relationship between guano enrichment and the aboveground biomass of vegetation. This suggests that while guano enhances nutrient dynamics and supports succession, its contribution to feedbacks between vegetation and landscape formation remains less certain under the studied conditions. One explanation may lie in the seasonal dynamics of trait expression: traits like biomass and height may temporarily increase due to spring nutrient uptake, enhancing sediment trapping (Zarnetske et al., 2012). However, this process may also reduce relative vegetation height as sediment accumulates (Derijckere et al., 2023). Since our measurements were taken in summer, the timing may have obscured any guano-related enhancement of biogeomorphic traits. Additionally, our approach of reducing dimensionality of plant traits, might also limited us to reveal the non-linear relationship between guano enrichment and bio geomorphic plant traits found in Reijers et al. (2024).

## 4.2 Guano as driver for landscape modification

Our remote sensing analysis confirms that seabird guano plays a significant role in shaping biogeomorphic dynamics across sandy islands in the Wadden Sea, particularly by enhancing sediment deposition in fertilised vegetated areas. We found that the interaction between guano and vegetation development differed depending on how the vegetation state change was measured. When vegetation dynamics were captured through early-season greening (ΔGI), guano amplified the positive effects of vegetation on sedimentation, indicating synergistic effects between fertilization and spring productivity. This suggests an earlier activation of biogeomorphic capabilities in plants when fertilised with guano (Figure 8). This supports the idea that plant ecosystem engineering is context-dependent and influenced by the timing and age-related expression of functional traits (Ven et al., 2023). A summer-season decline in vegetation cover compared to the previous year, particularly under guano-rich conditions, is associated with subsequent sediment burial (Figure 8). Notably, this pattern aligns with the observed spatial distribution of sedimentation on the western coast of Zuiderduin (Figure 7).

At this location, a sandy barrier, home to a dense cormorant colony and exposed to high rates of guano deposition, has accreted substantially. Due to rapid sediment burial, both NDVI values declined, reflecting a reduction in vegetation state in summer. The demise of vegetation cover is known to reflect sediment burial (Miller et al., 2010). However, both sediment trapping capacity and resilience to burial are closely tied to the vertical growth potential of biogeomorphic dune grasses (Strypsteen et al., 2024). While vegetation density may have decreased due to burial, it is possible that vertical growth was enhanced by guano fertilization, enabling the plants to maintain their role in sediment capture. Spatially explicit predictions reveal that in areas near seabird colonies, up to 22% of the positive sediment bed level change can be attributed to fertilization by guano.



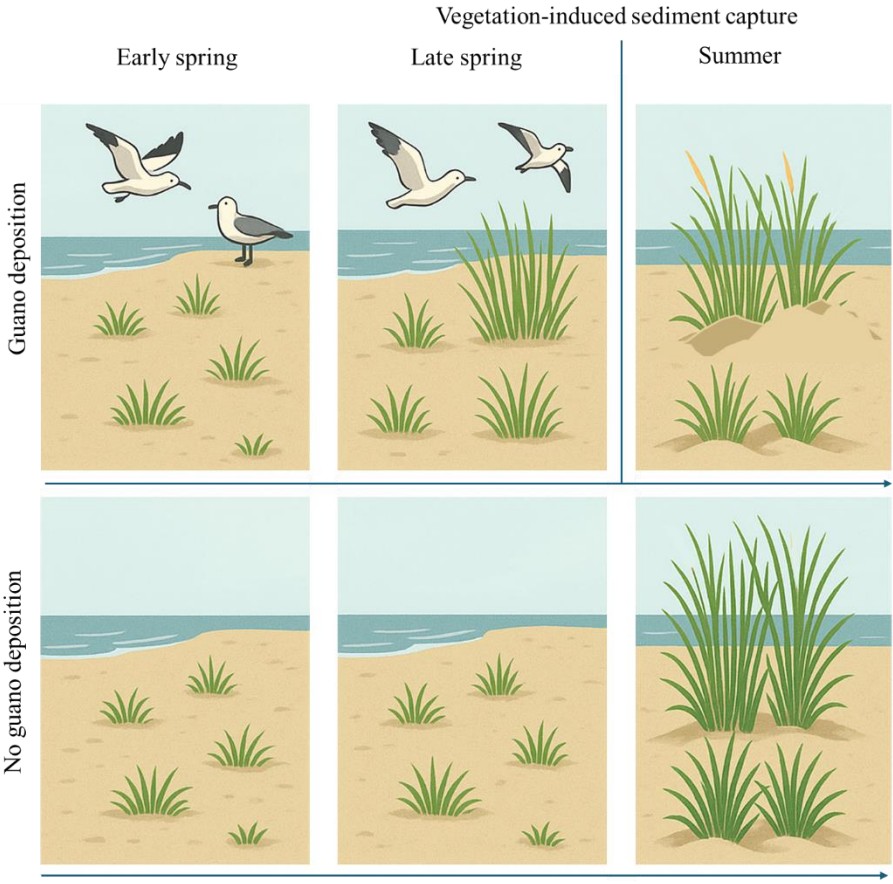

**Figure 8: This illustration compares vegetation development with and without guano deposition over time. In early spring (left column), guano input by seabirds initiates fertilization, enhancing plant growth. By spring (middle column), this leads to increased vegetation height and biomass, strengthening the plants' ability to trap sediment (biogeomorphic feedback). In summer (right column), this feedback is visible through dune formation in guano-enriched areas. However, the apparent vegetation height decreases slightly as plants become partially buried by sediment. In contrast, areas without guano show less growth and weaker biogeomorphic effects throughout the season.**

### 4.3 Mechanistic understanding and causality

This research contributes to the growing body of literature showing that allochthonous nutrient inputs, such as seabird guano, can influence plant species composition and functional traits (Ellis, 2005; Maron et al., 2006; Reijers et al., 2024). Crucially, we provide novel evidence that these bird-derived nutrient subsidies not only reshape vegetation dynamics but also drive landscape evolution, enhancing sediment deposition and coastal morphodynamics via strengthened vegetation–sediment feedbacks. Biophysical feedback mechanisms are well-documented in other coastal systems such as dunes (Bonte et al., 2021), salt marshes (Allen, 2000), seagrass beds (Forsberg et al., 2018), and mangroves (van Maanen et al., 2015).



Causality, however, remains difficult to establish. While our results show a positive association between guano deposition, vegetation dynamics, and sediment accumulation, these patterns could also reflect seabird preferences for nesting in areas that already have favorable conditions, such as stable, vegetated sites with higher sedimentation and less erosion (DeRose-Wilson et al., 2013; Raynor et al., 2012), where eggs are less likely to be washed away (Bailey et al., 2017; van de Pol et al., 2024b). It is likely that both processes, vegetation-induced sedimentation and seabird habitat selection, are simultaneously at play, suggesting the role of birds as agents of their own habitat formation. To disentangle these reciprocal relationships and better understand causality, manipulative experiments are needed that vary guano deposition rates and plant species identity to test how guano affects plant trait plasticity and sedimentation feedbacks. Yet, because these feedbacks unfold across spatial and temporal scales that are challenging to capture through fieldwork alone, empirical data must be integrated into trait-based ecological frameworks and dynamic numerical models. Currently, such models rarely incorporate plant traits, limiting their ability to simulate guano-driven geomorphological change. Advancing these modelling tools is critical, not only to improve our understanding of biogeomorphic processes but also to inform conservation strategies. As coastal ecosystems face growing threats from sea-level rise (van de Pol et al., 2024) and coastal squeeze (Lansu et al., 2024), technical progress is needed to predict how guano-mediated changes in vegetation influence landscape development and, in turn, seabird habitat availability (Paleczny et al., 2015). Ultimately, trait-informed models may help close the loop between vegetation dynamics and habitat selection, supporting more effective protection of coastal bird populations.

## 5 Conclusions

In conclusion, our spatially explicit study combines fine-scale field measurements of plant traits with landscape-scale remote sensing (NDVI and coastal elevation differencing) to provide the first quantitative evidence that seabird guano inputs modulate vegetation–sedimentation feedbacks and drive coastal landscape evolution. Structural equation models and Bayesian hierarchical remote-sensing analyses reveal that these guano effects on species composition, foliar nitrogen content, and sediment deposition are highly context-dependent, varying with elevation, substrate type, season, and island-specific conditions, nuances often overlooked in whole-island studies. Mechanistically, while high guano levels are associated with nitrophilous, high-productivity species and boosts foliar nitrogen, its impact on biogeomorphic traits such as biomass, rooting depth, and vegetation height exhibits pronounced spatial and seasonal variability. Our integrated approach, combining in situ trait measurements, structural equation modelling, remote sensing analyses, and Bayesian spatial modelling, provides a useful framework for exploring nutrient–vegetation–geomorphology interactions in other soft-sediment ecosystems.

*Data availability.* Upon acceptance, the data supporting our results will be freely available and the data DOI will be included.



*Author contributions.* FFR, LLG, CJC, GR and VCR conceptualized the study. FFR, MPAZ, CT collected all field data. CT and FFR performed all lab analyses. PG and FFR performed the remote sensing analysis. FFR performed all data analyses and wrote the original draft. VCR, CJC and LLG secured the funding. All authors reviewed on the original draft.

*Competing interests.* The contact author has declared that none of the authors has any competing interests.

*Acknowledgements.* We thank Ane Derk van Rees, Nadia Hijner, Solveig Hofer, Paul Berghuis, and Eva Lansu for their invaluable assistance during fieldwork. We are grateful to Wim-Jan Boon and all employees of the Waddenunit for their

support with transport to the islands and technical assistance, particularly Jan Kostwinner, Arjen Dijkstra, and Romke Kleefstra. We thank Richard Deen and Anne Dekkinga for their support for acquiring the permits for the fieldwork sites. For providing bird data and local ecological insights, we thank Jaap Kloosterhuis, Jan Veen, Erik Jansen, Allix Brenninkmeijer, Thea Smit, Marc van der Aa, Erwin Goutbeek, and Carl Zuhorn, and Kees Koffijberg.

*Financial support.* This work is part of the avian-nutrient pump project funded by a UU-NIOZ collaborative grand awarded to VCR and CJC. VCR was additionally supported by NWO-Veni grant VI.Veni.212.059.

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
