# Peer review of "Nutrient Flows and Biogeomorphic Feedbacks: Linking Seabird Guano to Plant traits and Morphological Change on Sandy Islands"

_EGUsphere, 2025_

## Referee Comment (RC1)

This manuscript offers insights into a field with limited existing research, with novelty in that they do not scale a small sampling area to a whole island.

Although the topic is slightly outside my primary area of expertise, the authors clearly articulate the need for their research and situate it well within the existing literature. Their methods are strong, and they avoid overinterpreting their results. The manuscript effectively outlines the current state of knowledge, identifies the gaps their study addresses, and highlights areas for future research.

There are, however, a few points of confusion that would benefit from clarification by the authors. These areas offer opportunities for improvement.

Below, specific comments are offered:

**Abstract/Introduction:**

Ln. 75-76 Why is it likely that the fertilizing impact is underestimated when extrapolating findings from a few square meters to the whole island? Would it not be overestimating if the few square meters were in an area near bird colonies, then extrapolating that high value to the whole island? Clarification could be used here. Additionally, this point could be tied back into the discussion. What were the values of this research compared to prior research? Did it seem that other papers underestimated or overestimated the impact of guano?

**Methods:**

Figure 1 On some of the islands, the MHW does not reach the vegetation border. Consider describing how this may influence sedimentation (e.g., if there is increased sediment delivery from the water).

Ln. 135 Can be more clear about plot level here, or leave off the (4 m2) until going into more detail in section 2.3. It feels confusing to bring up here – was it 4 m2 total per island? Per plot? How many plots? This comes up in section 2.2, but is read with confusion on Ln. 135. Additionally, it is confusing to say from the plot level on Ln. 135 then on Ln. 136, saying scaled up to the island level because the point of this research was not to scale up and extrapolate to the whole island level. After reading further, this made sense, but it could use more clarification here to avoid points of confusion when initially reading through.

Ln. 147. The range for the length of the transect, as well as the range for plots on a transect, are both reported, so it could be useful to report the range of transects within an island as well.

Ln. 149 Transects were placed to capture variation in guano deposition, both within and outside bird colonies, but how did transects vary by island morphology and hydrodynamics of the island? Statement in the introduction on Ln. 76-79 about how the effects of guano are heterogeneous, varying in both magnitude and direction, depending on local conditions such as topography and hydrodynamics. If transects were set based on guano deposition/colony location, are the plots also representative of local conditions?

Ln. 152 Could add a statement about how sampling during the breeding season influences results (e.g., more or less guano deposition during breeding season). Could also add here that this is the plant growing season, making this the optimal time for sampling.

Ln. 156 The sentence on rooting depth is separated from the sentence that says, "At each sampling plot," so consider adding this back in here. It is unclear whether two 1 m depth soil profiles were taken per plot, per transect, or per island.

Ln. 325 and 327 Could it be important to explicitly say slightly positive and slightly negative? These values both feel quite near 0. I feel some readers will skip over the values and simply read positive and negative.

**Results:**

Figure 7 caption, wrt may be common in some context, but this is a bit ambiguous. It's never introduced, and some readers may not know what it stands for. Additionally, the colonies line and >1.2 m elevation line are too similar in appearance, which gets particularly confusing for Rottumerplaat. A possible suggestion is adding hatching or stippling to the colonies to help differentiate. Transect lines could be added in a different color to the figure as well, but that may make it even busier.

Ln. 426 Curious what the range is, not just "up to," as this could make the impact seem larger than it is. (This is the same for line 373).

**Discussion:**

Any further discussion on how sand nourishment (nutrient-poor sand) at Griend could impact results? Mentioned in methods but not brought up again.

This research was done in the spring/summer, but there is never a mention of how these trends and influences may vary during different times of the year. Is there a long-term benefit to guano deposition with sediment accretion? Is there any benefit to guano deposition outside of the plant growing season? How may guano deposition vary outside of the breeding season?

**Technical Corrections:**

Ln. 48 The species name for Marram grass could be added.

Ln. 101 and Ln. 108 Inconsistency in describing the study islands as inhabited or uninhabited. I believe that Ln. 101 should say uninhabited.